# Maternal Dietary Patterns Are Associated with Pre-Pregnancy Body Mass Index and Gestational Weight Gain: Results from the “Mamma & Bambino” Cohort

**DOI:** 10.3390/nu11061308

**Published:** 2019-06-10

**Authors:** Andrea Maugeri, Martina Barchitta, Giuliana Favara, Maria Clara La Rosa, Claudia La Mastra, Roberta Magnano San Lio, Antonella Agodi

**Affiliations:** Department of Medical and Surgical Sciences and Advanced Technologies “GF Ingrassia”, University of Catania, Via S. Sofia 87, 95123 Catania, Italy; andreamaugeri88@gmail.com (A.M.); martina.barchitta@unict.it (M.B.); giuliana.favara@gmail.com (G.F.); mariclalarosa@gmail.com (M.C.L.R.); claudia.lamastra@libero.it (C.L.M.); robimagnano@gmail.com (R.M.S.L.)

**Keywords:** diet, pregnancy, dietary assessment, epidemiology, public health, neonatal outcomes

## Abstract

The present study investigated the association of maternal dietary patterns with pre-pregnancy body mass index (BMI) and total gestational weight gain (GWG), using data of 232 women from the “Mamma & Bambino” cohort. Dietary patterns were derived by a food frequency questionnaire and principal component analysis. Self-reported pre-pregnancy BMI and GWG were calculated according to the World Health Organization and Institute of Medicine guidelines, respectively. The adherence to the “Western” dietary pattern—characterized by high intake of red meat, fries, dipping sauces, salty snacks and alcoholic drinks—was associated with increased GWG (β = 1.217; standard error [SE] = 0.487; *p* = 0.013), especially among obese women (β = 7.363; SE = 1.808; *p* = 0.005). In contrast, the adherence to the “prudent” dietary pattern—characterized by high intake of boiled potatoes, cooked vegetables, legumes, pizza and soup—was associated with reduced pre-pregnancy BMI (β = −0.631; SE = 0.318; *p*-trend = 0.038). Interestingly, the adherence to this pattern was positively associated with GWG among underweight (β = 4.127; SE = 1.722; *p* = 0.048), and negatively among overweight and obese individuals (β = −4.209; SE = 1.635; *p* = 0.016 and β = −7.356; SE = 2.304; *p* = 0.031, respectively). Our findings point out that the promotion of a healthy diet might represent a potential preventive strategy against inadequate weight gain, even during the periconceptional period.

## 1. Introduction

Despite great efforts to tackle the obesity epidemic through public health policies and individual treatments, the World Health Organization (WHO) estimated that more than two billion adults worldwide were overweight or obese in 2014 [1]. The first thousand days of life, from conception to the end of the second year, represent the first critical period in the development of obesity. In early pregnancy, excessive maternal gestational weight gain (GWG) is a risk factor for increased birth weight, which it turn has been associated with a higher risk of obesity in childhood and adulthood [2,3]. Interestingly, a low birth weight has been also linked to higher body fat percentage and abdominal obesity in adolescents [4]. Beyond birth weight, the amount of GWG may affect both maternal and newborn health: in mothers, excessive GWG has been associated with an increased risk of hypertension [5], diabetes [6], cesarean section [7], postpartum weight retention [8] and obesity [9]; in newborns, the most common outcomes of inadequate GWG are neonatal and infant mortality, preterm birth and fetal growth retardation [10]. In 2009, to face the increasing burden of these adverse outcomes, the Institute of Medicine (IOM) revised the guidelines on recommended GWG, which took into account pre-pregnancy body mass index (BMI) independent of age and ethnicity [11]. Several lines of evidence suggest that only a third of mothers follow these recommendations, while more than half do not meet the IOM guidelines with adverse outcomes for mothers and newborns [9,12,13,14,15,16,17]. In general, potential preventive strategies for weight loss or controlling excessive weight gain should include promotion programs, addressing lifestyle behaviors and dietary habits [18]. Although evidence is currently limited, these interventions should start as early as possible, even during the periconceptional period. The study of dietary patterns during pregnancy is one of the suitable approaches to reveal the effect of diet on maternal and newborn health. Previous studies investigated the association of dietary patterns with pre-pregnancy BMI and/or GWG [19,20,21,22], but none of them has focused on populations from Southern Europe. Since dietary patterns vary across countries, it is important to identify country-specific dietary patterns that may be associated with the above-mentioned outcomes. To explore the effect of preconception, perinatal and early life exposure on maternal and newborn health, we designed the ongoing prospective “Mamma & Bambino” study, which enrolls mother–child pairs from Catania, Italy. Recently, we identified social determinants and lifestyles that affected quality of diet during pregnancy [23]. Our hypothesis was that dietary habits during the early phase of pregnancy—specifically those referred as “healthy”—might improve adequate GWG according to pre-gestational BMI. Thus, we used data from the “Mamma & Bambino” cohort to identify major maternal dietary patterns and to investigate their association with pre-pregnancy BMI and GWG.

## 2. Materials and Methods 

### 2.1. Study Design 

Study design and protocols of the “Mamma & Bambino” study are fully described elsewhere [23,24] and further information can be found at http://www.birthcohorts.net. In brief, pregnant women referred to the Azienda Ospedaliera Universitaria “Policlinico-Vittorio Emanuele” (Catania, Italy) were enrolled during prenatal genetic counseling, at 4–20 gestational weeks (mean = 16 weeks). A structured questionnaire was administered by trained epidemiologists to collect information on socio-demographic variables and lifestyle factors. Educational level was categorized as low-medium (primary school, i.e., ≤8 years of school) and high education level (high school education or greater, i.e., >8 years of school). Women were also classified as employed or unemployed (including students and housewives). Smoking status was classified as no-smoking (including ex-smokers) and currently smoking. We also collected data on the use of folic acid and other multivitamin supplements. In the current study, we included women from the “Mamma & Bambino” cohort who completed their pregnancy, while all the mothers with plurality, pre-existing medical conditions (i.e., autoimmune and/or chronic diseases), pregnancy complications (i.e., preeclampsia, gestational hypertension and diabetes), pre-term induced delivery or caesarean section, intrauterine fetal death and congenital malformations were excluded. The study protocol was approved by the ethics committee of the involved institution (CE Catania 2; Prot. N. 227/BE and 275/BE) and performed according to the Declaration of Helsinki. All women were fully informed of the purpose and procedures and gave written informed consent.

### 2.2. Definitions of Pre-Pregnancy Body Mass Index and Gestational Weight Gain

For the current investigation, self-reported maternal pre-pregnancy weight and height were collected at the recruitment. Pre-pregnancy BMI was calculated as weight in kilograms divided by height in meters squared and classified according to WHO criteria [25]. At the time of delivery, maternal weight, length of gestation, birth weight and birth length of newborns were collected from clinical records. Total GWG was calculated by subtracting the self-reported pre-pregnancy weight from the weight at delivery. According to the IOM guidelines, we defined adequate GWG as 12.5–18 kg for underweight, 11.5–16 kg for normal weight, 7–11.5 kg for overweight, and 5–9 kg for obese women [11]. 

### 2.3. Dietary Assessment

Dietary assessment was performed at the recruitment using a 95-item semi-quantitative Food Frequency Questionnaire (FFQ), which was referred to the previous month [26,27,28,29]. FFQ was adapted from a 46-item FFQ validated for the assessment of folate intake in Italian women of child-bearing age [26]. Since women were recruited at 4–20 weeks of gestation, dietary assessment was referred to as the early phase of pregnancy (i.e., from the beginning to the 16th week of gestation).

For each food item, women were asked to report the frequency of consumption and portion size: frequencies of food consumption were classified into twelve categories (i.e., from “almost never” to “two or more times a day”); portion sizes were classified into three categories (i.e., small, medium, and large), using an indicative photograph atlas. Diet information obtained from the FFQ were converted into monthly and daily food intakes, multiplying the frequency of consumption for portion size (g). Total energy intake was calculated using the table of food composition of the US Department of Agriculture (http://ndb.nal.usda.gov/), adapted to typical Italian food consumption. Food intakes were adjusted for total energy intake using the residual method [30].

### 2.4. Principal Component Analysis

Principal component analysis (PCA) was used to extract a posteriori dietary patterns, as described elsewhere [27,31]. In brief, we firstly classified the 95 FFQ food items into 39 predefined food groups, based on the similarity of nutrient profiles and culinary usage. Individual food items were preserved if they constituted a distinct item on their own (e.g., eggs, pizza, coffee or tea, etc.) or if they characterized a particular dietary pattern (e.g., wine, alcoholic drinks, and chips, etc.). Food group classification was based on previous studies conducted on women of child-bearing age, which used the same FFQ [23,27] (Appendix A). For each food group, the energy-adjusted value was entered into the factor analysis and factors were rotated by orthogonal transformation (varimax rotation). The number of retained dietary patterns was determined according to scree plot examination, eigenvalues >2.0), and interpretability. Factor loadings with an absolute value ≥0.2 were retained to define food groups that characterized each dietary pattern. For each dietary pattern, factor scores were calculated as the sum of products between observed energy-adjusted food group intakes and their factor loadings. According to factor loading distribution, adherence to each dietary pattern was categorized as low (1st tertile of factor loading), medium (2nd tertile), or high (3rd tertile). To corroborate internal reproducibility, factor analysis was separately replicated in two randomly selected subgroups (*n* = 100), using the same approach as for the main analysis. Next, we tested the correlation of factors scores between the overall sample and two randomly selected subgroups. Cohen’s weighted kappa was applied to assess the correct ranking ability by comparing tertile distribution in the overall cohort and in two randomly selected subgroups.

### 2.5. Statistical Analysis

Descriptive statistics were used to characterize the population using frequency or median and interquartile range (IQR). Prior to analysis, the normal distribution of all variables was checked using the Kolmogorov–Smirnov test. Continuous variables underlying skewed distribution were compared using the Kruskal–Wallis test for comparisons between three or more groups. Categorical variables were compared using the Chi-squared test. Linear regression models were used to investigate the association of dietary patterns with pre-pregnancy BMI and GWG, using the first tertile as reference. For each dietary pattern, we also evaluated the association of one-standard deviation increase in factor score with BMI and GWG. For pre-pregnancy BMI, the model was adjusted for age, educational level, employment status, smoking, total energy intake and gestational age at recruitment. We also tested for interaction between gestational age at recruitment and adherence to dietary pattern on pregestational BMI. For GWG, linear regression models were run on the overall population and stratified by pre-pregnancy BMI categories. The model adjusted for variables meant to be associated with GWG, including age, length of gestation, birth weight, educational level, working status, smoking, parity, newborn sex and total energy intake. Statistical analyses were performed using SPSS software (version 22.0, SPSS, Chicago, IL, USA). All statistical tests were two-sided, and *p* values < 0.05 were considered statistically significant.

## 3. Results

### 3.1. Characteristics of Study Population

Overall, 232 women who completed singleton pregnancy (aged 15–50 years, median = 37 years) were enrolled in the “Mamma & Bambino” cohort, at a median gestational age of 16 weeks. Based on pre-gestational BMI, we identified 8.1% underweight, 66.2% normal weight, 16.7% overweight and 9% obese women. According to pre-gestational BMI and GWG, we identified 31.2% and 27.4% who exhibited reduced or excessive GWG, respectively. Maternal characteristics according to GWG categories are showed in Table 1. In general, we observed U-shaped distributions of pre-pregnancy weight and BMI across GWG categories (*p* < 0.001 and *p* = 0.002, respectively). In fact, women with adequate GWG reported lower pre-pregnancy weight and BMI compared to those with reduced or excessive GWG. Moreover, both maternal weight at delivery and newborn birth weight were higher in women with excessive GWG compared to those with adequate or reduced GWG (*p* < 0.001 and *p* = 0.039, respectively).

### 3.2. Dietary Patterns and Pre-Pregnancy Body Mass Index

According to Scree plot examination (Appendix A), we first derived two dietary patterns with eigenvalues >2.0, explaining 15.55% of the total variance among 39 food groups. Other PCA components which explained less than 5.0% of total variance were excluded. The radar graph illustrates factor loadings which characterized each dietary pattern in the whole cohort (Figure 1), while factor loadings obtained in two randomly selected groups are shown in Appendix A. The analysis of dietary patterns in two randomly selected subgroups yielded similar results. Indeed, factor scores obtained in the subgroups were well correlated with those obtained in the whole cohort (Spearman’s correlation coefficient ranged from 0.8 to 0.9). Moreover, we demonstrated almost perfect agreement in the ranking ability between the whole cohort and two randomly selected subgroups (weighted kappa from 0.81 to 1.0).

In the whole cohort, the “Western” dietary pattern was characterized by high intake of red meat, fries, dipping sauces, salty snacks and alcoholic drinks. The higher adherence to the Western dietary pattern was associated with decreasing age and percentage of high-educated women (*p* < 0.001 and *p* = 0.022, respectively) (Table 2). 

In contrast, the “prudent” dietary pattern was characterized by a high intake of boiled potatoes, cooked vegetables, legumes, pizza and soup. Particularly, pre-pregnancy weight and BMI decreased across tertiles of the prudent dietary pattern (*p* = 0.043 and *p* = 0.019, respectively). In fact, women with high adherence to the prudent dietary pattern were less likely to be overweight or obese (*p* = 0.007) (Table 3). Results of the linear regression confirmed the negative association between pre-pregnancy BMI and the adherence to the prudent dietary pattern, after adjustment for age, educational level, employment status, smoking, total energy intake and gestational age at recruitment (β = −0.631; SE = 0.318; *p*-trend = 0.038). Particularly, women in the 3rd tertile of the prudent dietary pattern exhibited a ~1.4 point reduced pre-pregnancy BMI, compared to those in the 1st tertile (β = −1.347; SE = 0.598; *p* = 0.024). No association between pre-pregnancy BMI and the adherence to the Western dietary pattern, nor interaction with gestational age at recruitment were evident.

### 3.3. Dietary Patterns and Gestational Weight Gain

Although no associations between dietary patterns and GWG were found in the univariate analysis, we performed a linear regression model adjusting for age, weight at delivery, gestational duration, educational level, working, smoking, parity, newborn sex and total energy intake (Table 4). In general, we reported a positive trend of GWG across tertiles of the Western dietary pattern (β = 1.217; SE = 0.487; *p* = 0.013). A similar trend was observed in obese women (β = 7.363; SE = 1.808; *p* = 0.005), with women in the 3rd tertile exhibiting a ~13.7 Kg increased GWG compared to those in the 1st tertile (β = 13.701; SE = 0.887; *p* = 0.041). By contrast, no association between GWG and adherence to the prudent dietary pattern was evident in the overall population. However, we observed an opposite effect across BMI categories: among underweight, we found a positive trend of GWG across tertiles of the prudent dietary pattern (β = 4.127; SE = 1.722; *p* = 0.048); conversely, we showed a negative trend in overweight (β = −4.209; SE = 1.635; *p* = 0.016) and obese (β = −7.356; SE = 2.304; *p* = 0.031) women. Particularly, overweight women in the 2nd and 3rd tertiles exhibited a ~8.0 and ~9.8 Kg reduced GWG, respectively, compared to those in the 1st tertile (β = −7.975; SE = 2.672; *p* = 0.010; and β = −9.736; SE = 4.302; *p* = 0.037).

## 4. Discussion

To our knowledge, the present study is the first to investigate the association of maternal dietary patterns with pre-pregnancy BMI and GWG in Southern Europe. The novelty of our study was that we demonstrated how “healthy” dietary habits in the early phase of pregnancy might promote adequate GWG according to pre-gestational BMI. Indeed, among mothers of the “Mamma & Bambino” cohort (Catania, Italy), we identified two dietary patterns: the prudent dietary pattern was characterized by a high intake of boiled potatoes, cooked vegetables, legumes, pizza and soup; by contrast, the Western dietary pattern was characterized by a high intake of red meat, fries, dipping sauces, salty snacks and alcoholic drinks. We first demonstrated that adherence to the prudent dietary pattern was negatively associated with pre-pregnancy BMI. Indeed, women with high adherence reported ~1.3 point reduced pre-pregnancy BMI, compared to those with low adherence. Interestingly, we demonstrated a dual opposite effect of the prudent dietary pattern on GWG across BMI categories: the adherence to this pattern was positively associated with GWG among underweight women, and negatively associated among overweight and obese women. Particularly, compared to low adherents, overweight women with medium and high adherence to the prudent dietary pattern exhibited ~8.0Kg and ~9.8 Kg reduced GWG, respectively. Although some studies failed in demonstrating the association of healthy diet with GWG [19,20], results from the prospective Norwegian Mother and Child cohort study reported that underweight women with high adherence to the New Nordic Diet—characterized by a high intake of fruits and vegetables—had lower risk of excessive GWG [22]. These results and ours might be attributable to the healthy effect of fruits and vegetables on women’s health and fetal growth. For instance, a prospective study of US pregnant women reported significantly lower GWG among women who consumed more than three servings/day of fruits and vegetables compared with <3 servings/day [32]. Similarly, a cross-sectional analysis within the National Health and Nutrition Examination Survey (NHANES) project demonstrated greater odds of exceeding GWG guidelines in pregnant women who consumed low amounts of total vegetables than those who consumed higher amounts [33]. The adherence to a prudent dietary pattern might provide a balanced intake of energy, macro- and micronutrients [32] and, therefore, promote an adequate GWG independent of pregestational BMI. Indeed, foods rich in vitamins, minerals, fibers, and antioxidants can help stimulate the immune system and detoxification enzymes, improve cholesterol synthesis, modulate hormone metabolism and stimulate antioxidant defense [34]. Moreover, women who adhered to a prudent diet were more likely to have a healthy lifestyle, which in turn contributed to adequate GWG independent of pre-pregnancy BMI [35]. However, further research is encouraged to evaluate the effects of certain food categories and the potential mechanistic pathways involved in this process. The dual effect across BMI categories might be also partially explained by dietary reporting errors [36] and/or in different composition of gestational weight gain [37]—which consists of gains in total body water, fat-free mass and fat mass—across BMI categories. Thus, our study demonstrated that adherence to a prudent dietary pattern ameliorated pre-pregnancy BMI and GWG across BMI categories, which in turn has been previously associated with increased birth weight [38]. Of note, our study also confirmed that birthweight was higher in infants born from mothers with excessive GWG, compared to those with adequate or reduced GWG.

With respect to the Western dietary pattern, we showed that pregnant women with high adherence were more likely to be younger and less educated. This is in line with evidence that older women have a healthy lifestyle in general [22] and that healthy food choices reflect an overall healthy behavior [39,40]. Contrary to the prudent dietary pattern, adherence to the Western dietary pattern was not associated with pre-gestational BMI. This raises the need of recommendations on foods to be consumed (e.g., fruit and vegetables) instead of on foods to be avoided during pregnancy. However, we demonstrated a positive association between the adherence to the Western dietary pattern and GWG, in general and especially in obese women. Particularly, among the latter, high adherence to the Western dietary pattern (those in the 3rd tertile) led to a ~13.7 Kg increased GWG compared to the low adherence (1st tertile). Although few studies evaluated the association of maternal unhealthy diet with GWG [19,20,21,22,41,42], our findings are consistent with previous studies demonstrating that dietary patterns characterized by junk foods, salty snacks, high-fat and high-sugar foods were associated with higher GWG [19,20]. Particularly, Tielemans et al. found a positive association of adherence to different dietary patterns—named “Margarine, sugar and snacks” and “Vegetable, oil and fish”, respectively—with higher GWG [41]. A plausible explanation is that the Western dietary pattern is characterized by unhealthy and energy-dense foods—specifically fries, dipping sauces, salty snacks and alcoholic drinks —which could increase total energy intake. Indeed, a recent systematic review demonstrated a positive association between energy intake and GWG [43]. 

Our study had some limitations. The cross-sectional design precluded assessing causality of the association between dietary patterns and pre-pregnancy BMI. Thus, we cannot completely exclude a potential reverse causation in the observed relationship. To manage this issue, we adjusted our analysis for gestational age at recruitment and tested its interaction with adherence to dietary patterns on pregestational BMI. Notably, no interaction was evident suggesting that our results were independent of gestational age at recruitment. With respect to GWG, we evaluated the difference between weight before pregnancy and at delivery, instead of data at each trimester. This did not allow us to investigate the effect of dietary patterns on GWG trajectories. Although self-reported weight assessment restricted the power of our work, it has been previously established that self-reported pre-pregnancy weight correlates well with measured pre-pregnancy weight [44,45]. However, in our study, reporting errors of weight cannot be completely excluded and further research should evaluate components of GWG, including total body water, fat-free mass and fat mass. Moreover, due to the limited sample size in underweight and obese groups, we were not able to evaluate the effect of adherence to dietary patterns on adequate GWG, assessed as dichotomous outcomes. Indeed, for dichotomous outcomes, the rule of thumb suggests using one independent variable for ten subjects. By contrast, the required number of subjects for variable for linear regression appears much smaller than in logistic regression [46]. Indeed, Austin and Steyerberg demonstrated that two subjects per variable tends to permit accurate estimation of regression coefficients in a linear regression model [46]. With respect to dietary assessment, we recognized that PCA-derived dietary patterns only explained 15.55% of total variance among food groups. However, we used well-established criteria (i.e., scree plot examination, eigenvalues >2.0 and interpretability) to derive dietary patterns that, therefore, were consistent with those reported by previous studies conducted on similar populations [27,31]. Furthermore, since dietary assessment referred to the early phase of pregnancy (i.e., from the beginning to 16 weeks of gestation), we were not able to account for changes in maternal dietary pattern. Nevertheless, it has been demonstrated that dietary patterns may not change largely during pregnancy despite an increased energy intake [47,48]. Despite this evidence, changes in some dietary habits during pregnancy—especially those related to alcoholic drinks—cannot be completely excluded [49]. Accordingly, previous studies excluded the alcohol component from their dietary assessment [21]. However, in our cohort, the intake of alcoholic drinks characterized the Western dietary pattern, underlining that some women did not follow the recommendations on avoiding alcohol intake during pregnancy. Finally, we cannot rule out the possibility of bias from residual confounders that might affect both maternal dietary patterns and GWG, such as physical activity, sedentary lifestyle, unmeasured socio-demographic factors, psychosocial conditions, illness and vomiting. Notably, Tielemans and colleagues demonstrated that the exclusion of women who vomited more than once per week did not alter their results [41]. Overall, these limitations discourage generalizability of our findings to other populations and settings. However, in our opinion, the analysis of dietary patterns represents a practical and useful approach to inform and to guide healthy eating and weight management during pregnancy. Accordingly, the prudent dietary pattern—including a variety of healthy foods—has been encouraged in many dietary guidelines [35].

## 5. Conclusions

Our study was the first investigating the association of dietary patterns with pre-pregnancy BMI and GWG in women from Southern Europe. We demonstrated that a Western dietary pattern did not affect pre-gestational BMI but increased GWG, especially in obese women. In contrast, a prudent dietary pattern rich in potatoes, cooked vegetables, legumes, pizza and soup, ameliorated pre-gestational BMI and GWG, with different effects across BMI categories. Indeed, the adherence to this pattern was positively associated with GWG among underweight women, and negatively associated among overweight and obese women. Thus, the promotion of healthy dietary habits, even during the periconceptional period, represents a potential strategy to maintain an adequate weight independent of pre-gestational BMI.

## Figures and Tables

**Figure 1 nutrients-11-01308-f001:**
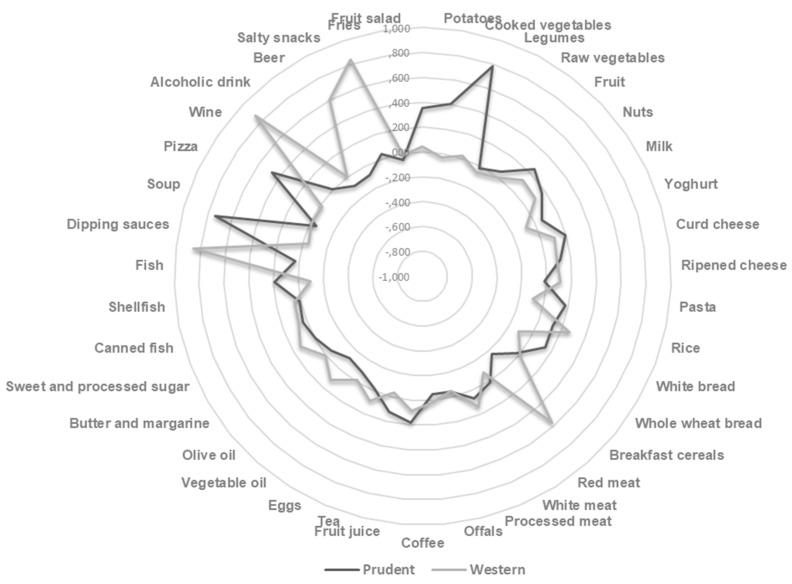
Radar graph of factor loadings that characterize each dietary pattern.

**Table 1 nutrients-11-01308-t001:** Characteristics of women from the “Mamma & Bambino” cohort (*n* = 232) according to gestational weight gain categories.

Characteristics	Reduced GWG (*n* = 73)	Adequate GWG (*n* = 95)	Excessive GWG (*n* = 64)	*p*-Value ^a^
**Age ^b^**	37.0 (4.0)	38.0 (5.0)	37.0 (4.0)	0.546
**Educational level (low-medium %) ^c^**	15.1%	13.4%	17.2%	0.804
**Working (%)**	58.9%	63.9%	54.7%	0.495
**Smoking (%)**	15.1%	17.7%	22.2%	0.553
**Pre-pregnancy weight ^b^**	61.0 (13.3)	59.0 (13.0)	64.0 (17.8)	<0.001
**Pre-pregnancy BMI ^b^**	23.1 (4.4)	21.6 (3.8)	24.2 (6.5)	0.002
**Pre-pregnancy BMI categories**				
Underweight	6.8%	8.2%	9.4%	0.001
Normal weight	68.5%	77.3%	46.9%	
Overweight	15.1%	7.2%	32.8%	
Obese	9.6%	7.2%	10.9%	
**Weight at delivery ^b^**	68.0 (10.0)	72.0 (14.0)	82.0 (16.7)	<0.001
**Length of gestation ^b^**	39.0 (2.0)	39.0 (2.0)	39.2 (2.0)	0.701
**Birth weight ^b^**	3.2 (0.6)	3.2 (0.6)	3.3 (0.6)	0.039
**Birth length ^b^**	50.0 (2.0)	50.0 (2.0)	50.0 (2.0)	0.286

^a^*p*-values are based on the Kruskal–Wallis test for quantitative variables, or Chi-squared test for categorical variables; ^b^ data are reported as median interquartile range (IQR); ^c^ defined as ≤8 years of school. Abbreviations: GWG, gestational weight gain; BMI, body mass index.

**Table 2 nutrients-11-01308-t002:** Characteristics of women from the “Mamma & Bambino” cohort (*n* = 232) according to adherence to the Western dietary pattern.

Characteristics	1st Tertile	2nd Tertile	3rd Tertile	*p*-Value ^a^
**Age ^b^**	38.0 (5.0)	38.0 (4.0)	36.0 (3.0)	<0.001
**Gestational age ^b^**	16.0 (3.0)	16.0 (4.0)	16.0 (2.0)	0.777
**Educational level (low-medium %) ^c^**	20.0%	11.7%	26.4%	0.022
**Working (%)**	59.1%	63.1%	52.7%	0.291
**Smoking (%)**	17.4%	16.2%	27.5%	0.074
**Use of folic acid supplements (%)**	95.1%	94.7%	94.7%	0.949
**Use of other multivitamin supplements (%)**	44.4%	33.3%	42.1%	0.334
**Pre-pregnancy weight ^b^**	60.0 (14.2)	62.5 (15.0)	60.0 (15.0)	0.923
**Pre-pregnancy BMI ^b^**	22.3 (4.4)	22.7 (5.0)	22.8 (5.5)	0.704
**Pre-pregnancy BMI categories**				
Underweight	7.3%	7.2%	6.4%	0.687
Normal weight	63.6%	69.4%	66.1%	
Overweight	17.3%	13.5%	21.1%	
Obese	11.8%	9.9%	6.4%	
**Weight at delivery ^b^**	71.5 (16.5)	74.0 (16.0)	74.0 (14.0)	0.636
**Length of gestation ^b^**	39.0 (2.0)	39.0 (2.0)	39.0 (2.0)	0.976
**Birth weight ^b^**	3.2 (0.6)	3.2 (0.7)	3.3 (0.5)	0.800
**Length ^b^**	50.0 (2.0)	50.0 (1.0)	50.0 (2.0)	0.391
**GWG ^b^**	11.5 (7.2)	13.0 (7.0)	13.0 (9.0)	0.056
**GWG classification**				
Reduced	36.6%	28.0%	28.9%	0.162
Adequate	41.5%	48.0%	34.2%	
Excessive	22%	24%	36.8%	

^a^*p*-values are based on the Kruskal–Wallis test for quantitative variables, or Chi-squared test for categorical variables; ^b^ data are reported as median (IQR); ^c^ defined as ≤8 years of school. Abbreviations: GWG, gestational weight gain; BMI, body mass index.

**Table 3 nutrients-11-01308-t003:** Characteristics of women from the “Mamma & Bambino” cohort (*n* = 232) according to adherence to the prudent dietary pattern.

Characteristics	1st Tertile	2nd Tertile	3rd Tertile	*p*-Value ^a^
**Age ^b^**	38.0 (5.0)	37.0 (4.0)	37.0 (4.0)	0.675
**Gestational age ^b^**	16.0 (1.0)	16.0 (3.0)	15.0 (5.0)	0.001
**Educational level (low-medium %) ^c^**	22.7%	19.8%	15.5%	0.389
**Working (%)**	57.3%	60.4%	57.3%	0.865
**Smoking (%)**	20.9%	18.0%	22.2%	0.731
**Use of folic acid supplements (%)**	91.7%	93.8%	98.7%	0.210
**Use of other multivitamin supplements (%)**	59.7%	61.0%	59.0%	0.966
**Pre-pregnancy weight ^b^**	63.0 (12.0)	60.5 (14.2)	58.5 (14.0)	0.043
**Pre-pregnancy BMI ^b^**	23.2 (4.7)	22.7 (4.7)	21.8 (5.1)	0.019
**Pre-pregnancy BMI categories**				
Underweight	5.5%	8.1%	7.3%	0.007
Normal weight	64.5%	65.8%	70.8%	
Overweight	20.9%	17.4%	14.5%	
Obese	9.1%	8.7%	7.4%	
**Weight at delivery ^b^**	74.0 (17.0)	73.5 (14.2)	72.0 (15.0)	0.551
**Length of gestation ^b^**	39.0 (2.0)	39.0 (2.0)	39.0 (2.0)	0.562
**Birth weight ^b^**	3.2 (0.6)	3.2 (0.6)	3.3 (0.7)	0.522
**Length ^b^**	50.0 (2.0)	50.0 (2.0)	50.0 (2.0)	0.935
**GWG ^b^**	12.0 (8.0)	12.0 (6.2)	13.0 (7.5)	0.830
**GWG classification**				
Reduced	31.9%	34.1%	27.8%	0.823
Adequate	37.5%	40.2%	45.6%	
Excessive	30.6%	25.6%	26.6%	

^a^*p*-values are based on the Kruskal–Wallis test for quantitative variables, or Chi-squared test for categorical variables; ^b^ data are reported as median (IQR); ^c^ defined as ≤8 years of school. Abbreviations: GWG, gestational weight gain; BMI, body mass index.

**Table 4 nutrients-11-01308-t004:** Linear regression of the association between dietary patterns and gestational weight gain, stratified by body mass index categories.

Dietary Patterns	Total			Underweight			Normal Weight			Overweight			Obese		
	β	SE	*p*-Value	β	SE	*p*-Value	β	SE	*p*-Value	β	SE	*p*-Value	β	SE	*p*-Value
**Western**															
**1st tertile**	Ref			Ref			Ref			Ref			Ref		
**2nd tertile**	1.369	0.971	0.161	1.198	5.516	0.848	1.218	0.992	0.223	5.003	4.152	0.250	2.549	3.967	0.636
**3rd tertile**	1.542	1.072	0.152	2.308	10.321	0.860	0.961	1.116	0.392	1.917	3.637	0.605	13.701	0.887	**0.041**
**Trend**	1.217	0.487	**0.013**	−0.425	1.651	0.804	0.372	0.542	0.493	2.695	1.828	0.152	7.363	1.808	**0.005**
**Prudent**															
**1st tertile**	Ref			Ref			Ref			Ref			Ref		
**2nd tertile**	−0.353	1.019	0.730	−5.149	1.351	0.163	0.895	1.098	0.417	−7.975	2.672	**0.010**	−5.730	2.156	0.131
**3rd tertile**	0.184	1.067	0.863	5.382	1.678	0.274	−0.003	1.142	0.998	−9.736	4.302	**0.037**	−10.730	4.156	0.061
**Trend**	0.118	0.513	0.818	4.127	1.722	**0.048**	0.046	0.538	0.932	−4.209	1.635	**0.016**	−7.356	2.304	**0.031**

The model was adjusted for age, length of gestation, birth weight, educational level, working status, smoking, parity, newborn sex and total energy intake. *p*-values < 0.05 are indicated in bold font. Abbreviations: SE, standard error; Ref, reference group.

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
