# Peer review of "Maternal Dietary Patterns Are Associated with Pre-Pregnancy Body Mass Index and Gestational Weight Gain: Results from the “Mamma & Bambino” Cohort"

_nutrients, 2019, doi:10.3390/nu11061308_

Round 1
Reviewer 1 Report
This study, from the “Mamma & Bambino” cohort, examines the relationship between pre-pregnancy maternal dietary patterns and pre-pregnancy BMI and gestational weight gain. This study finds that a healthy or “prudent” dietary pattern is positively associated with lower pre-pregnancy BMI, and negatively associated with GWG in overweight and obese women. These results contribute to our understanding of dietary habits that may promote appropriate weight gain during pregnancy. While interesting, there are major limitations and questions regarding the categorization and descriptions of dietary patterns and timing of measurements that need to be addressed.
Major
1. This study assessed dietary patterns pre-pregnancy, but no assessments during pregnancy . The authors recognize the major limitation that this study does not account for changes in maternal dietary pattern across pregnancy is a major limitation. The authors note that dietary pattens may not change during pregnancy however this conclusion is particularly concerning for alcoholic drinks, wine and beer. FFQ was assessed in the month prior to pregnancy, therefore consumption of alcohol may be inaccurately reflected in this study. Previously studies actually removed alcohol from model of dietary pattern (Rifas-Shiman, 2009) and alcohol intake does change pre and during pregnancy (Inksip, BMJ, 2009)
2. The two dietary patterns only explain 15.5% of total variance among 39 food groups. This seems relatively low in term of overall dietary pattern. Please explain that this total variance is sufficient to derive actual dietary habits.
3. Although the methods state that the 39 food groups based on similarity of nutrient profiles or culinary usage, however it is not clear why certain foods/groups were not included together. For example, why was milk, yoghurt, curd cheese and ripened cheese not categorized as dairy? Similarly, for a pregnancy cohort, why were alcoholic drinks, beer and wine separated? Please explain in greater detail the rationale for grouping foods.
4. The results indicated that pre-pregnancy prudent dietary pattern was positively associated with GWG in underweight women, and negatively among overweight and obese women. This is an interesting observation that warrants further discussion. Please include the influence of dietary pattern on GWG classification, for example did following a prudent dietary pattern help with gaining a recommended amount of weight (based on pre-pregnancy BMI category)? The association between dietary patterns and appropriate gestational weight gain based on BMI classifications is an important consideration.
5. The description of dietary patterns is extremely vague and does not provide much context for other researchers. For example dipping sauces, salty snacks – what foods does this category include, and how was this determined in the FFQ?
6. In the prudent dietary pattern, potatoes were included, and in the western dietary pattern, fries were included. Did the FFQ ask specific questions regarding preparation of potatoes to allow for these categorizations. Were all potatoes (except for fries) included as part of the prudent dietary pattern?
7. Results indicated that adherence to prudent dietary pattern ameliorated pre-pregnancy BMI and GWG across BMI categories, but no association with western dietary pattern. Please discuss further as this an interesting observation.
8. Did the analysis of dietary patterns include lower intakes of western dietary pattern in the prudent dietary pattern, ie lower intake of sugary foods, red meat, and vice versa for western dietary pattern, low intake of V&F? Lower consumption of certain food categories based on dietary pattern may be as important as high consumption of other food categories.
Minor
1. The authors note that none of previous studies focused on populations from Southern Europe and it is important to identify country-specific dietary patterns. Instead of focusing on interventions, the authors should discuss what is novel about this study other than just reporting in a different population.
2. The study objective was to investigate the association of maternal dietary patterns with pre-pregnancy BMI and GWG, but where is the hypothesis?
3. Methods mention that the FFQ conducted using the previous month as the reference period – so was this the month before enrollment, pre-pregnancy?
4. Was supplement intake assessed in this cohort, and if so, did this factor into dietary patterns or adjusted at all?
5. The authors mentioned internal reproducibility was validated using two subgroups. Please include this validation in a supplementary table.
6. In the PCA – explain what categorized by tertiles means, is this low, medium and high adherence? Was this scored?
7. Previous study examining social determined and lifestyles and diet quality during pregnancy had n=332 in cohort, this study only has n=232. Were the n=100 women excluded for a particular reason, or just timing of the analysis? Please explain the reason for the discrepancy.
8. Table 2 – what is the summarized p-value for pre-pregnancy BMI categories and GWG referring to? Should include a footnote in both Table 2 and 3 on what the p-values indicate
9. Table 3 – explain what low-medium educational level
Author Response
Dear Editor,
thank you very much for considering our manuscript and for comments of independent Reviewers. We submit to your attention a revised version of the manuscript in which we have considered all comments. The following List of change and answers to comments of Reviewers addresses all changes made in the manuscript (red font).
List of change and answers to comments of Reviewers
Reviewer 1
This study, from the “Mamma & Bambino” cohort, examines the relationship between pre-pregnancy maternal dietary patterns and pre-pregnancy BMI and gestational weight gain. This study finds that a healthy or “prudent” dietary pattern is positively associated with lower pre-pregnancy BMI, and negatively associated with GWG in overweight and obese women. These results contribute to our understanding of dietary habits that may promote appropriate weight gain during pregnancy. While interesting, there are major limitations and questions regarding the categorization and descriptions of dietary patterns and timing of measurements that need to be addressed.
We are very grateful with Reviewer 1 for his/her positive comments on out manuscript. We believe that his/her suggestions helped us in improving our manuscript.
Major
1. This study assessed dietary patterns pre-pregnancy, but no assessments during pregnancy. The authors recognize the major limitation that this study does not account for changes in maternal dietary pattern across pregnancy is a major limitation. The authors note that dietary patterns may not change during pregnancy however this conclusion is particularly concerning for alcoholic drinks, wine and beer. FFQ was assessed in the month prior to pregnancy, therefore consumption of alcohol may be inaccurately reflected in this study. Previously studies actually removed alcohol from model of dietary pattern (Rifas-Shiman, 2009) and alcohol intake does change pre and during pregnancy (Inksip, BMJ, 2009).
We are grateful with Reviewer 1 for this comment. Our hypothesis was that dietary patterns at the beginning of pregnancy might be associated with pre-gestational BMI and GWG. To investigate this hypothesis, we evaluated maternal dietary patterns in the early phase of pregnancy and not in the month prior to pregnancy (i.e. we enrolled women from the 4 to the 20 weeks of gestation and FFQ was the referred to the previous month from the enrollment). Our results demonstrated that alcoholic drinks positively characterized the western dietary pattern in the first trimester of pregnancy, and that increasing adherence to this dietary pattern was associated with increasing GWG. According to Reviewer 1 comment, we recognize that drinking pattern might change during pregnancy and we included it as a limitation of our study, also citing the proposed references (lines 326-331).
2. The two dietary patterns only explain 15.5% of total variance among 39 food groups. This seems relatively low in term of overall dietary pattern. Please explain that this total variance is sufficient to derive actual dietary habits.
We are grateful with Reviewer 1 for his/her helpful comment. In our study, we used Principal component analysis (PCA) to extract a posteriori dietary patterns, and the number of retained dietary patterns was determined according to scree plot examination, eigenvalues >2.0, and interpretability. Based on these criteria, we selected two dietary patterns that explained 15.55% of total variance among 39 food groups. We recognized that the explained variance is too low, but, in our opinion, we used a set of well-established criteria for determining the number of patterns to be retained. Indeed, dietary patterns derived in our study are consistent with those reported in previous studies conducted on similar populations. Moreover, to corroborate internal reproducibility, factor analysis was separately replicated in two randomly selected subgroups, using the same approach as for the main analysis (lines 119-124 and 167-173). Given that, we believe that total variance is sufficient to derive actual dietary patterns in the Mamma & Bambino cohort, and that dietary patterns are consistent with those identified in the same cohort and/or in other studies. Anyway, according to the suggestion, we included the low explained variance as a limitation of our study (lines 319-323).
3. Although the methods state that the 39 food groups based on similarity of nutrient profiles or culinary usage, however it is not clear why certain foods/groups were not included together. For example, why was milk, yoghurt, curd cheese and ripened cheese not categorized as dairy? Similarly, for a pregnancy cohort, why were alcoholic drinks, beer and wine separated? Please explain in greater detail the rationale for grouping foods.
As described in the method section, food groups were classified based on nutrient profiles and culinary usage. For instance, milk, yoghurt, curd cheese and ripened cheese are considered as dairy products but they have different nutrient profile. Moreover, they are differently consumed during the day (i.e. milk and yoghurt at breakfast while cheese at lunch or dinner). However, to improve the comprehension of the classification of food groups we provided the Supplementary table 1.
4. The results indicated that pre-pregnancy prudent dietary pattern was positively associated with GWG in underweight women, and negatively among overweight and obese women. This is an interesting observation that warrants further discussion. Please include the influence of dietary pattern on GWG classification, for example did following a prudent dietary pattern help with gaining a recommended amount of weight (based on pre-pregnancy BMI category)? The association between dietary patterns and appropriate gestational weight gain based on BMI classifications is an important consideration.
We agree with Reviewer 1 that our results, especially those related to GWG, warrant further discussion (please see the revised version of the discussion section). However, due to the limited sample size, we cannot assess the effect of dietary pattern on GWG classification. Indeed, for dichotomous outcomes, the rule of thumb is “one independent variable for ten subjects”. The limited numbers of underweight and obese women do not allow us to perform logistic regression analysis. By contrast, the required number of subjects for variable for linear regression appears much smaller than in logistic or Cox regression. Indeed, Austin and Steyerberg demonstrated that two subjects per variable tends to permit accurate estimation of regression coefficients in a linear regression model (lines 313-319). Based on this evidence, we prefer to maintain data stratified by weight status using the linear regression model.
5. The description of dietary patterns is extremely vague and does not provide much context for other researchers. For example, dipping sauces, salty snacks – what foods does this category include, and how was this determined in the FFQ?
We are sorry if description of dietary patterns was unclear. To improve the comprehension of the classification of food groups and the meaning of dietary patterns we provided the Supplementary table 1.
6. In the prudent dietary pattern, potatoes were included, and in the western dietary pattern, fries were included. Did the FFQ ask specific questions regarding preparation of potatoes to allow for these categorizations. Were all potatoes (except for fries) included as part of the prudent dietary pattern?
We used a FFQ that separately asked on potatoes (boiled potatoes) and fries. In our study, boiled potatoes positively characterized the prudent dietary pattern while fries positively characterized the western dietary pattern. However, according to Reviewer comment, we included the Supplementary table 1 to improve the comprehension of food groups.
7. Results indicated that adherence to prudent dietary pattern ameliorated pre-pregnancy BMI and GWG across BMI categories, but no association with western dietary pattern. Please discuss further as this an interesting observation.
We are grateful with Reviewer 1 for his/her positive comments on our interesting results. To give more emphasis on the different effects of dietary patterns on BMI and GWG, we have revised the discussion and the conclusion sections.
8. Did the analysis of dietary patterns include lower intakes of western dietary pattern in the prudent dietary pattern, ie lower intake of sugary foods, red meat, and vice versa for western dietary pattern, low intake of V&F? Lower consumption of certain food categories based on dietary pattern may be as important as high consumption of other food categories.
We agree with Reviewer 1 that lower consumption of certain food categories might be as important as high consumption of other food categories. However, our approach based on PCA is directed to evaluate the effect of dietary patterns instead of single food categories. Anyway, derivation of dietary patterns takes into account all the 39 food groups. This means that women who adhered more to the prudent dietary pattern might consume little amount of “western” foods and vice versa. To support the request of Reviewer 1, we discussed in the discussion section the potential effects of specific food groups, and we encouraged further research which takes into account the effect of specific food groups.
Minor
1. The authors note that none of previous studies focused on populations from Southern Europe and it is important to identify country-specific dietary patterns. Instead of focusing on interventions, the authors should discuss what is novel about this study other than just reporting in a different population.
According to the Reviewer suggestion, we added the novelty of our study in the discussion section (lines 245-247). In our opinion, the novelty of our study was that we demonstrated how “healthy” dietary habits in the early phase of pregnancy might promote adequate GWG according to pre-gestational BMI.
2. The study objective was to investigate the association of maternal dietary patterns with pre-pregnancy BMI and GWG, but where is the hypothesis?
According to Reviewer suggestion, we included our hypothesis at the end of the introduction section (lines 56-58).
3. Methods mention that the FFQ conducted using the previous month as the reference period – so was this the month before enrollment, pre-pregnancy?
We are sorry if details on dietary assessment are unclear. As reported in this revised version of our manuscript, dietary assessment was performed at the recruitment using a FFQ referred to the previous month. Since women were enrolled from 4 to 20 gestational week, dietary assessment is referred to the early phase of pregnancy (i.e. from the beginning to the 16 week of gestation) (lines 92-96).
4. Was supplement intake assessed in this cohort, and if so, did this factor into dietary patterns or adjusted at all?
The use of supplements (folic acid or other multivitamin supplements) was assessed in the Mamma & Bambino cohort. Accordingly, we added the collection of this information in the method section (line 72) and we provided the percentage of users in tables 1-3. However, no difference by gestational weight gain and adherence to dietary patterns was evident.
5. The authors mentioned internal reproducibility was validated using two subgroups. Please include this validation in a supplementary table.
According to the Reviewer suggestion, we added more details on the reproducibility of dietary assessment in two randomly selected groups. Particularly, we included the Supplementary Table 2 that displays factor loadings obtained in two subgroups, and results from correlation test and Cohen Kappa analysis (lines 168-173).
6. In the PCA – explain what categorized by tertiles means, is this low, medium and high adherence? Was this scored?
According to this comment, we better explained that adherence to each dietary pattern was evaluated according to factor loading distribution, and categorized as low (1st tertile of factor loading), medium (2nd tertile), or high (3rd tertile) (lines 118-119).
7. Previous study examining social determined and lifestyles and diet quality during pregnancy had n=332 in cohort, this study only has n=232. Were the n=100 women excluded for a particular reason, or just timing of the analysis? Please explain the reason for the discrepancy.
We are sorry for the discrepancy. However in the previous study, which examined the effect of social determinants on diet quality, we included all the women enrolled by the Mamma & Bambino cohort. Instead, in the current study, we included only women who completed their pregnancy to assess their gestational weight gain. Anyway, we better explained inclusion/exclusion criteria in the study design section (lines 72-74).
8. Table 2 – what is the summarized p-value for pre-pregnancy BMI categories and GWG referring to? Should include a footnote in both Table 2 and 3 on what the p-values indicate
As suggested, we included a footnote explaining statistical analysis for p-value computation.
9. Table 3 – explain what low-medium educational level
As suggested, we explained that low-medium educational level was defined as ≤8 years of school
Reviewer 2 Report
Summary
The aim of this study was investigate prospectively the association of maternal dietary patterns with pre-pregnancy BMI and total gestational weight gain (GWG). Data from 232 women from the “Mamma & Bambino” cohort were used. Dietary patterns were derived by principal component analysis. The adherence to the “Western” dietary pattern was associated with increased GWG. In contrast, the adherence to the “prudent! Dietary patterns was associated with reduced pre-pregnancy BMI.
The manuscript is well written and sophisticated analysis are done. However, some issues need to be addressed.to strengthen this report.
Major points
Materials and methods
1. Page 3, line 97: The grouping into 39 items, is this based on previous research/articles that used a similar FFQ? This process needs a bit more explanation if it’s not based on any previously done research.
Results
2. Page 4, Line 138: why is an eigenvalue of >2 used? As this explains only 15.6% of the total variance, which is extremely low. Why are not lower eigenvalues used, to increase the variance?
3. Page 5, Lines 148-149: rice is not that characteristic according to figure 1 for the prudent diet, pizza would be more characteristically according to the cohort’s data. Can the authors comment and elaborate on the choice for rice over pizza?
4. Page 6: what are the exact numbers for women who are underweight, normal weight, overweight and obese? If a rule of thumb of one confounder for every ten women is used, the number of confounders is probably too large for these subgroup analyses. This may lead to overfitting, which may lead to regression coefficients explaining noise rather than real relationships between GWG and dietary patterns and also might reduce generalisability of the results. Suggest not to show the data stratified by weight status.
Discussion
5. In the discussion is stated that dietary assessment was done on the first month of pregnancy. However, as the women were recruited between 4-20 weeks (mean=16 wks), and the FFQ had to be filled out using the previous month as a reference period, how could this be the first month of pregnancy? And if that is the case, might morning sickness have played a role in filling out the FFQ then? Women with morning sickness do adjust their diet during that period. Can the authors elaborate a bit on this?
Minor points
Materials and methods
6. Page 2, lines 82-83: Please describe how the GWG status was determined using the guidelines.
7. Page 2, section 2.3: Was the FFQ validated? If not, how was it developed? Based on another validated tool?
Results
8. Page 4, Line 143: in this sentence, the readers are referred to table 2, but that should be table 3. Table 2 is on prudent diet. And vice versa for line 152 on page 5.
9. Reference 11 is not a reference to the IOM guidelines. Please include the correct reference.
10. Tables: Table titles should include more information, suggest to include the name of the cohort and the number of women included.
Author Response
Dear Editor,
thank you very much for considering our manuscript and for comments of independent Reviewers. We submit to your attention a revised version of the manuscript in which we have considered all comments. The following List of change and answers to comments of Reviewers addresses all changes made in the manuscript (red font).
List of change and answers to comments of Reviewers
Reviewer 2
Summary
The aim of this study was investigate prospectively the association of maternal dietary patterns with pre-pregnancy BMI and total gestational weight gain (GWG). Data from 232 women from the “Mamma & Bambino” cohort were used. Dietary patterns were derived by principal component analysis. The adherence to the “Western” dietary pattern was associated with increased GWG. In contrast, the adherence to the “prudent! Dietary patterns was associated with reduced pre-pregnancy BMI.
The manuscript is well written and sophisticated analysis are done. However, some issues need to be addressed.to strengthen this report.
We are very grateful with Reviewer 2 for his/her positive comments on out manuscript. We believe that his/her suggestions helped us in improving our manuscript.
Major points
Materials and methods
1. Page 3, line 97: The grouping into 39 items, is this based on previous research/articles that used a similar FFQ? This process needs a bit more explanation if it’s not based on any previously done research.
According to Reviewer 2 suggestion, we better explained that food group classification was based on previous studies conducted on women of child-bearing age with the same FFQ (lines 92-96). However, to improve the understanding of food group classification we provided the Supplementary table 1.
Results
2. Page 4, Line 138: why is an eigenvalue of >2 used? As this explains only 15.6% of the total variance, which is extremely low. Why are not lower eigenvalues used, to increase the variance?
In our study, we used Principal component analysis (PCA) to extract a posteriori dietary patterns. The number of retained dietary patterns was determined according to scree plot examination, eigenvalues >2.0, and interpretability. Based on these criteria, we selected two dietary patterns that explained 15.55% of total variance among 39 food groups. We recognized that the explained variance is too low, but, in our opinion, we used a set of well-established criteria for determining the number of patterns to be retained. In the new version of our manuscript, we also provided the Scree plot (Supplementary figure 1) which display eigenvalue for each PCA component. As reported in the results section, other PCA components (i.e. potential dietary patterns) explained less than 5% of total variance (lines 165-166). Accordingly, we retained only the first two dietary patterns. Dietary patterns derived in our study are consistent with those reported in previous studies conducted on similar populations. Moreover, to corroborate internal reproducibility, factor analysis was separately replicated in two randomly selected subgroups, using the same approach as for the main analysis. Factor loadings obtained in two randomly selected groups are shown in Supplementary Table 2. Notably, the analysis of dietary patterns in two randomly selected subgroups yielded similar results. Indeed, factor scores obtained in the subgroups well correlated with those obtained in the whole cohort (Spearman’s correlation coefficient ranged from 0.8 to 0.9). Moreover, we demonstrated almost perfect agreement in the ranking ability between the whole cohort and two randomly selected subgroups (weighted kappa from 0.81 to 1.0) (lines 168-173). Given that, we believe that total variance is sufficient to derive actual dietary patterns in the Mamma & Bambino cohort, and that dietary patterns are consistent with those identified in the same cohort and/or in other studies. Anyway, according to the suggestion, we included the low explained variance as a limitation of our study (lines 319-323).
3. Page 5, Lines 148-149: rice is not that characteristic according to figure 1 for the prudent diet, pizza would be more characteristically according to the cohort’s data. Can the authors comment and elaborate on the choice for rice over pizza?
We are very grateful with Reviewer 2 for his/her comment and we are sorry for our mistake. Factor loadings with an absolute value ≥0.2 were retained to define food groups that characterized each dietary pattern. Accordingly, we have revised the description of dietary patterns. Indeed, as correctly suggested, the prudent dietary pattern was characterized by high intake of boiled potatoes, cooked vegetables, legumes, pizza and soup.
4. Page 6: what are the exact numbers for women who are underweight, normal weight, overweight and obese? If a rule of thumb of one confounder for every ten women is used, the number of confounders is probably too large for these subgroup analyses. This may lead to overfitting, which may lead to regression coefficients explaining noise rather than real relationships between GWG and dietary patterns and also might reduce generalisability of the results. Suggest not to show the data stratified by weight status.
As suggested by Reviewer 2, we included the percentage of women according to their pre-gestational BMI in the result section (lines 145-147). In the context of dichotomous outcomes, we agree with the rule of thumb of one independent variable for ten subjects. By contrast, the required number of subject for variable for linear regression appears much smaller than in logistic or Cox regression. Indeed, Austin and Steyerberg demonstrated that two subjects per variable tends to permit accurate estimation of regression coefficients in a linear regression model (lines 313-319). Based on this evidence, we prefer to maintain data stratified by weight status.
Discussion
5. In the discussion is stated that dietary assessment was done on the first month of pregnancy. However, as the women were recruited between 4-20 weeks (mean=16 wks), and the FFQ had to be filled out using the previous month as a reference period, how could this be the first month of pregnancy? And if that is the case, might morning sickness have played a role in filling out the FFQ then? Women with morning sickness do adjust their diet during that period. Can the authors elaborate a bit on this?
We are sorry if definition of timing of dietary assessment is quite confusing. As reported in the revised version of our manuscript (lines 92-96), dietary assessment was performed at the recruitment using the previous month as reference period. Since women were recruited at 4-20 week of gestation, dietary assessment was referred to the early phase of pregnancy (i.e. from the beginning to the 16 week of gestation). We also revised this statement in the discussion section accordingly. Moreover, we elaborated on the potential effect of morning sickness also citing an appropriate reference (lines 334-335).
Minor points
Materials and methods
6. Page 2, lines 82-83: Please describe how the GWG status was determined using the guidelines.
As suggested, we better described how we determined adequate GWG based on the IoM guidelines (lines 87-89).
7. Page 2, section 2.3: Was the FFQ validated? If not, how was it developed? Based on another validated tool?
According to this suggestion, we provided more details about FFQ and its validation/development (lines 92-96).
Results
8. Page 4, Line 143: in this sentence, the readers are referred to table 2, but that should be table 3. Table 2 is on prudent diet. And vice versa for line 152 on page 5.
We are sorry for this mistake. According to Reviewer suggestion, we revised the order of tables 2 and 3.
9. Reference 11 is not a reference to the IOM guidelines. Please include the correct reference.
We are sorry for the mistake. We corrected the reference accordingly.
10. Tables: Table titles should include more information, suggest to include the name of the cohort and the number of women included.
As suggested, we included the name of the cohort and the number of included women in table titles.
Round 2
Reviewer 1 Report
The authors have diligently addressed comments note by this reviewer and have significantly improved the quality and clarity of the manuscript. The addition of Supplementary Tables 1 and 2 provide further details regarding the food categories which aids in the interpretation of this manuscript. All comments have been addressed to satisfaction.
Reviewer 2 Report
In my opinion, the rebuttal is clear, all comments are carefully and thoughtfully addressed and changes are clearly highlighted by the author. I have no further comments.